# Indicator from the graph Laplacian of stock market time series cross-sections can precisely determine the durations of market crashes

**Zheng Tien Kang**, **Peter Tsung-Wen Yen**, **Siew Ann Cheong** *

Division of Physics and Applied Physics, School of Physical and Mathematical Sciences, Nanyang Technological University, Singapore, Republic of Singapore

* cheongsa@ntu.edu.sg

**Data availability statement:** All data and scripts, less those we do not have ownership over, can be found at the repository: https://doi.org/10.21979/N9/7YNZAQ.

## Abstract

Stock market crashes are believed to occur unpredictably and have profound negative impacts on the economy and society. However, there is no universally agreed-upon definition of stock market crashes, whether it is an actual market state (implying that there is a start and an end) or just a transition between two different states (implying that it is a point event). Conventionally, extreme events in the financial markets can be determined using various change-point detection methods. However, these methods typically rely on a model of the time series data and/or use sliding time windows. Expanding on our previous work, we propose an alternative way of defining market crashes as short states by utilizing information cross-filtering by two time windows of the time derivative of the maximum Laplacian spectral gap across filtration parameters $1.0 \leq \epsilon \leq 1.8$. When we applied this method to analyze the time derivative of the maximum spectral gap for S&P 500, Nikkei 225, SGX and TWSE, we found persistent peaks (found across different time window widths) associated with the COVID-19 crash starting in March 2020 and ending only in April 2020. These dates correspond roughly with the highest point before the crash and the lowest point after the crash seen in the indices. We also found non-persistent peaks (found only across short time windows or long time windows) before and after the COVID-19 crash. The explanations for these non-persistent peaks are peculiar to individual markets, and also particular market crashes such as the 2008 Global Financial Crisis. Based on this work, we argue that a definition of market crash in terms of a duration is more natural and perhaps more useful for risk management.

## Introduction

Market crashes have been common over the last 10 years. As seen in Table 1, since 2007, we have witnessed at least 10 market crash events worldwide, not including those on a smaller scale. A common assumption is that the daily market returns are normally distributed, with a mean close to zero, and standard deviation between 1%–2% (see Table 1 in Ref [1]). If this is true, a simple Monte Carlo simulation would show us that a market event more negative

**Funding:** This study was supported by Ministry of Education, Singapore (Grant number MOE-T2EP20222-0017 to SAC). The funder did not play any role in the study design, data collection and analysis, decision to publish, or preparation of the manuscript.

**Competing interests:** The authors have declared that no competing interests exist.

than –5% would occur once every 10,000 years or so. The actual distribution is closer to being a Levy stable distribution with fat tails [2]. Market crashes are seemingly unpredictable [3–5], although there is increasing evidence that early warning signals precede them [6–9], just like what happens for critical transitions [10,11]. More importantly, market crashes are commonly treated as point events, even by econophysicists who have experience identifying market states. There is at present no universally accepted definition of a market crash, including whether it is a point event or an event with finite duration.

In the Literature review section, we survey previous works by econophysicists on self-organized market states, how many there are, and what are their characteristics. Indeed, most econophysicists agree that market crashes form a state of their own, and are well-separated from the other market states. We then show the dearth of previous works specifically targeted at determining the durations of market crashes. Following this, we survey existing methods for detecting the starts and ends of market crashes. These methods are collectively known as change point detection methods. We discuss their strengths and weaknesses when using these to detect market crashes with finite durations, before identifying the key knowledge gaps.

The key problem that motivated this study is the lack of temporal precision in distinguishing the market crash state from other market states in the financial time series data. To convincingly argue that a sudden change in the financial time series corresponds to a market crash, it is common to start from the cross-correlation matrix [12–14]. Unfortunately, the computation of cross correlations requires us to use a time window. This is what Zheng et al. [15] did in 2012, where they computed cross-correlation matrices, and used their changes in terms of principal component analysis (PCA) as indicators of systemic risks (including market crashes). Thereafter, in 2018, Pharasi et al. [16] utilized the cross-correlation structure to infer the possible characteristics of a market crash. Clearly, we cannot time the start and end of a market crash down to the day when we use a time window of several months to compute the cross-correlations.

In a recent paper on the Laplacian spectra of cross-correlation matrices [17], we found that there is no gap in the spectrum of a time window if it covers only the normal market phase, but a gap emerges whenever the time window overlaps with a market crash. When we examine the sequence of sliding time windows, we found that the change from a gapless spectrum to a gapped spectrum can be very sharp, and is associated with the time window just incorporating the market crash or just missing the market crash. Our contribution in this paper is thus a method of using two time windows to explore this highly sensitive indicator to determine the starts and ends of market crashes. After the Literature Review section, we proceed

**Table 1. In this table, we list major worldwide stock market crashes from 2007 to 2023. For each crash, we show its name, rough time of occurrence, stock index's high and low, and in which country it occurred.**

| Crash Event | Start Date | High | Low | Country |
|---|---|---|---|---|
| Financial Crisis of 2007-2008 | 16 Sep 2008 | 1561 | 683 | US |
| 2009 Dubai Debt standstill | 27 Nov 2009 | | | UAE |
| European Sovereign Debt Crisis | 27 Apr 2010 | | | Europe |
| 2010 Flash Crash | 6 May 2010 | 10460 | 9870 | US |
| August 2011 Stock Market Fall | 1 Aug 2011 | 1370 | 1074 | US |
| 2015-2016 Chinese Stock Market Crash | 12 Jun 2015 | 4478 | 2860 | China |
| 2015-2016 Stock Market Selloff | 18 Aug 2015 | 18312 | 17504 | US |
| 2018 Cryptocurrency Crash | 20 Sep 2018 | 19783 | 5500 | |
| 2020 Stock Market Crash | 24 Feb 2020 | 3386 | 2237 | |
| 2022 Stock Market Crash | 3 Jan 2022 | 4796 | 3666 | |
| 2022 Russian Stock Market Crash | 16 Feb 2022 | 4261 | 1944 | Russia |

to describe the data used and how it is processed in the Materials and Methods section. In this same section, we explain how our method of pairing two time windows works, before applying the method to real-world stock market data in the Results section. We discuss the strengths and limitations of our method, the implications of our findings in the Discussion section and conclude in the Conclusion section.

## Literature review

In this section, let us review past research works on market phases/states. Intuitively, it is natural to associate market phases with low and high volatility periods. The Great Moderation period in the US from the 1980s till the onset of the Global Financial Crisis is characterized by declining volatility on real stock returns, albeit with intermittent shocks such as the 1987 Black Monday incident and the 2001 September 11 crisis [18]. Over this long period, the market in a highly stable growth phase made occasional and brief transitions to the market crash state.

Economists were the first to classify financial time series data into two market states according to their volatilities. The first such attempt was by Hamilton [19], while other early attempts include those by Schwert [20] and Turner, Startz, and Nelson [21]. In these previous works, the Markov switching method was used to distinguish between the two states of the macroeconomy by analyzing different economic and financial time series. Stock returns data were later incorporated into such models in Schaller and Van Norden [22] and Baustista's works [23]. The first attempt by an econophysicist to classify market states was by Marsili, who applied the maximum likelihood clustering technique to identify six different market states in the 2000 New York Stock Exchange (NYSE) daily stock return data from 1 January 1990 to 30 April 1999 [24]. Each state is characterized by the movements of different industries and economic sectors in the economy. Other econophysicists then follow suit [12–14,16]. Our research group has also found six to seven market states in the US [25,26] and Japanese markets [27].

In addition, we surveyed past literature measuring the duration of market crashes. Cochran and Defina first measured the duration dependence of stock market cycles using a parametric hazard model in 1995 and found that post-World War II (WWII) contractions exhibit duration dependence while those prior to WWII do not [28]. This is followed by drawdown analysis of market crashes lasting a few days by Sornette and Johansen in 2001 where they studied the 14 largest drawdowns of the Dow Jones Industrial Average (DJIA) in this century to detect the presence of dragon kings from the probability distributions of drawdowns [29]. They applied the same drawdown analysis to detect dragon kings in the historical drawdown distributions of other financial assets such as currencies (e.g., DM/US$, Yen/US$), stock indices (e.g., S&P 500, NASDAQ Composite, Nikkei 225, DAX, FTSE 100, Hang Seng Index) and treasury bonds (US and Japanese) and commodities (Gold) in a 2010 follow-up paper [30]. Finally, in a 2017 paper, Barroa and Ursúa validated the positive relationship between macroeconomic changes and stock returns across different durations of several years [31]. They found that periods of economic depression are marked by frequent market crashes.

If indeed a market crash is a brief change in phase in the financial time series between two time points (i.e., having a start time and end time), we can detect it using change point analysis [32–34]. In general, there are three classes of methods: (1) hidden Markov models [19,20, 35,36], (2) dynamic programming [37,38], (3) entropic segmentation [39,40]. However, these methods typically apply to one time series at a time. If we apply them across a time series cross-section, the locations of change points will be different for different time series [25–27].

It is possible to detect market-wide change points by treating the cross-section of scalar time series as a single vector time series [41–43], or using a multivariate change point detection method [44,45].

Analyzing market phase/state changes from the perspective of a change-point problem poses its own problems. Because a market crash is treated as the combination of two successive change points, we might find the start but not the end, or the end but not the start, or get only one of them correct. We would love to show examples of these in financial time series data, but could not find any. Therefore, we review the gene-detection literature in bioinformatics instead [46–53]. A gene is a subsequence in the DNA that codes for a protein. Therefore, we expect an initial point (start codon) and an end point (stop codon) which delimits the gene from non-coding regions of the DNA. In prokaryotic genomes, very small genes are frequently missed, and we might also be confused by overlapping Open Reading Frames on opposite DNA strands (potential candidates for genes and their 'shadows') [48]. In eukaryotes, the gene structures are more complicated because there is an additional splicing stage between transcription and translation [48,49]. This introduces ambiguity in the gene detection process, because a second gene may start before the first gene ends. This is a problem that might also occur in stock markets, whereby a market crash occurs within a longer period of market corrections. The contemporary literature on gene detection focuses on the use of hidden Markov Models, dynamic programming, Support Vector Machines (SVM), Self-organized Mapping (SOM) and more recently, neural networks [48–52,54]. It is also possible for a CPD algorithm to detect the start codon but not the end codon, or the end but not the start, or incorrect pairs where both the start and end positions are wrong [49,51–53]. Overall, gene detection methods published usually have success rates of 80%, meaning that 20% of the genes are missed. We expect to encounter similar limitations when dealing with the identification of stock market phase changes.

Associated with the change in market phase/state is the problem of early warning indicators. In particular, we find many previous works predicting market crashes through the use of Schefferian early warning indicators (SEWIs). SEWIs include increased variance, skewness, and kurtosis, as well as critical slowing down and spectral reddening, along with others such as flickering and patchiness [55–59]. In their reviews [10,11], Scheffer et al. explained how the dynamics of a complex system slow down as it approaches a critical point or tipping point, taking longer to return to its equilibrium after perturbations. This universal behavior precedes all critical transitions; thus, the SEWIs can be used to identify and predict crashes in financial markets. Inspired by these works, we also studied the US housing market crash during 2004 to 2008 [60,61], and more recently extreme events in high-frequency FOREX time series [7], using the SEWIs.

Another popular approach to predicting market crashes is the log-periodic power law (LPPL) method, which is also used by many groups. This method was developed independently by Feigenbaum et al. [62,63] and Sornette et al. [64]. Through fitting the financial time series to power laws with log-periodic decorations, we can estimate how far away the crash (or any other extreme events) is from the current time, the degree of super-exponential growth of the current market state, and the angular frequency of the log-periodicity [65–73]. It is important to note that both the SEWI and LPPL methods provide insights on individual time series, neglecting cross-correlational changes among stocks.

Enlightened by the thorough literature review above, we identified two key gaps in our knowledge. First, there is the lack of suitable change point detection methods to detect market crashes as a state with finite duration in a single step. Most change point detection algorithms identify this duration by detecting the start and the end in two steps. Only the hidden Markov Model is capable of detecting the market crash as a separate hidden state in one step, but they

are computationally very expensive. This explains why we use the cross-filtering method as a response to the primary research question (i.e., to find the market crash state in one step). As a secondary research question, we also want to know whether there are noticeable differences when we apply our method to developed markets versus emerging markets.

MSCI, which is an authoritative source in the finance industry, lists Hong Kong and Singapore as developed markets [74] (see pages 4 and 5 of the table in the link). However, we disagree with the classification of Singapore because traders on the Singapore Exchange (SGX) do not have the same level of sophistication as those in the NYSE and the Tokyo Stock Exchange (TSE). Also, company valuations on the SGX are lower than similar companies listed on other markets. Overall, it is a much more illiquid market compared to the former two and has seen substantial delisting/privatization far outpacing new IPOs since the late 2010s [75–79]. Our assessment of Singapore as an emerging market is corroborated by a recent study, showing that Singapore shares similar emerging market attributes with other developing ASEAN markets [80].

We also reviewed works from various Economics and Finance journals, noting that different researchers have different opinions on the taxonomy of stock markets. Firstly, through clustering and centrality analysis of the minimal spanning tree (MST) of world stock market indices, it is found that throughout 2005 to 2014 US and the developed European markets are closely linked with one another. Emerging Asian markets are further apart and linked to the developed markets via the Japanese market. The developed Japanese market, together with the US and German markets, act as hubs in the network of world stock markets [81]. A separate dynamic correlation analysis suggested that Hong Kong and Japan both exhibited developed market characteristics [82]. The US market is generally regarded as the engine of growth for the whole world, thus it is natural to include it in our studies. We also chose the Japanese market as a contrasting developed market, because it is in Asia and because the GDP of Japan is the third largest (formerly second largest) in the world.

Having included the Japanese market as one of the two developed markets, it is natural to include emerging markets from Asia as a contrast. In particular, the Asian Tigers of Hong Kong, Singapore, South Korea, and Taiwan would be very good choices because of their rapid ascent as emerging markets. In an analysis of events unfolding during the 2008 US Subprime Crisis, South Korea and Singapore are considered emerging markets [83]. In another study investigating the financial interdependence between markets, South Korea and Taiwan are listed alongside Mexico as among the more advanced emerging markets [80]. However, we realized that it is not necessary to include all four markets in our study, not least because of the classification of Hong Kong as a developed market. In general, strong correlations are observed between the Hong Kong and Singapore markets, as seen during the 1997 Asian Financial Crisis [82] and their relative positions in the MST analysis of the recent COVID-19 crash [81,84,85]. Due to their similar industry mix and focus on the semiconductor industry, we also see that Taiwan is the nearest neighbor of South Korea in the MST [81,84,85]. Therefore, we chose Singapore from the Hong Kong-Singapore pair and Taiwan from the South Korea-Taiwan pair of more advanced emerging markets as contrasts to the US and Japanese markets.

## Materials and methods

### Data

Yahoo! Finance (https://finance.yahoo.com/) is a popular source for daily time series data for stocks in markets worldwide, because of its ease of use and its reasonable terms and conditions. Users interested in a few stocks or indices can visit the website, search for these

stocks or indices, and download historical data through a simple web interface. For many more stocks, this downloading process would have to be automated, typically through the use of scripts that would bypass the web interface, so long as the terms and conditions listed on https://legal.yahoo.com/us/en/yahoo/terms/product-atos/apiforydn/index.html are respected. Several open-source scripts are available, but we used Python's pandas datareader module to download from Yahoo! Finance the daily prices of all the component stocks in the Standard & Poor's 500 (S&P 500) index, the Nikkei 225 index, all listed stocks on the Singapore Stock Exchange (SGX), all listed stocks on the Taiwan Stock Exchange (TWSE) from 1 Jan 2019 to 30 Jun 2022.

Over the years, the component stocks of S&P 500 and Nikkei 225, as well as stocks listed in SGX and TWSE changed because some stocks were delisted and other stocks were newly listed. In the US markets, a stock retains its ticker symbol when the name of the company is changed, whereas in the Japanese markets a new ticker symbol is assigned to a stock even for name changes. From Yahoo! Finance, we can only download historical prices of ticker symbols that were active at the time of the data acquisition. For SGX, this problem was so severe that we had to manually download the historical prices of 75 stocks from https://www.investing.com. No problems were encountered with the TWSE. The total cross sections we managed to download were 505 (former/present) component stocks of the S&P 500, 220 (former/present) component stocks of the Nikkei 225, 630 stocks from the SGX, and 1020 stocks from the TWSE. These time series cross-section data were organized into four pandas dataframes that can be saved to Python pickle files using the `to_pickle()` function, and read by the `read_pickle()` function for subsequent analyses. Our method for collecting the data and subsequent analysis of this data complied with the terms and conditions of Yahoo! Finance. To answer the secondary research question, we selected the developed markets S&P 500 and Nikkei 225 to contrast the emerging markets SGX and TWSE. This choice allows us to compare the spectral features of all four markets to see if they are universal, or there are differences between developed and emerging markets.

For these data sets to be useful, we need to perform some pre-processing. First, the ticker symbols in each market were downloaded from Yahoo! Finance. Within our period of study, some stocks have missing price values and thus when we computed the fractional returns using these price values, we ended up with 'NaN's whenever one of the prices is missing. We then replaced the 'NaN's with '0's, and removed stocks with more than 50% '0's in their return time series.

## Cross correlations

For all the four markets, we computed the Pearson cross correlations

$$C_{ij} = \frac{\frac{1}{N_k} \sum_{t=1}^{N_k} (r_{i,t} - \bar{r}_i)(r_{j,t} - \bar{r}_j)}{\sqrt{\frac{1}{N_k-1} \sum_{t=1}^{N_k} (r_{i,t} - \bar{r}_i)^2} \sqrt{\frac{1}{N_k-1} \sum_{t=1}^{N_k} (r_{j,t} - \bar{r}_j)^2}} \tag{1}$$

of the daily price returns $r_{i,t}$ and $r_{j,t}$ within $k$-month time windows with $N_k + 1$ trading days. We then advanced the fixed $k$-month time window ($k = 2, 2.5, 3, 3.5, 4, 4.5, 5, 5.5,$ and $6$) starting from 1 January 2019. Here, $\bar{r}_i$ and $\bar{r}_j$ are the average returns of stocks $i$ and $j$ within the $k$-month time windows. For each time window, we converted the pairwise cross-correlations $C_{ij}$ into pairwise ultrametric distances $0 \leq d_{ij} = \sqrt{2(1 - C_{ij})} \leq 2$.

## Maximum spectral gap indicator

A common way to visualize the cross-correlations between $N$ stocks is to treat each stock as a node and draw a link between node $i$ and node $j$ if $C_{ij}$ is greater than a threshold $C_0$ that we choose. This network of $N$ stocks can be described by an adjacency matrix $A_{ij} = 1$ if nodes $i$ and $j$ are linked ($A_{ij} = 0$ otherwise), or in terms of the graph Laplacian $L_{ij} = K_{ij} - A_{ij}$ where $K_{ij}$ is the diagonal degree matrix ($K_{ii} = \sum_{j=1}^{N} A_{ij}$). If we increase $C_0$ from $C_0 = -1$ to $C_0 = 1$, we obtain a sequence of networks with fewer and fewer links. This filtration process shown schematically in Fig 1 is normally done by increasing a length scale $\epsilon$ from $\epsilon = 0$ to $\epsilon = 2$, so that nodes $i$ and $j$ are linked if $d_{ij} < \epsilon$.

If we obtain the eigenvalues $\{\lambda_k\}_{k=1}^{N}$ by diagonalizing $L_{ij}$, we can define the maximum spectral gap as maximum $\Delta\lambda = \max_{k=2}^{N-1}(\lambda_{k+1} - \lambda_k)$. In our recent work [17], we showed that maximum $\Delta\lambda$ is large during a market crash, and maximum $\Delta\lambda = 0$ when the market is in its normal phase. If we diagonalize the graph Laplacian $L$ in Fig 1(a), we would obtain four zero eigenvalues $\lambda_1 = \lambda_2 = \lambda_3 = \lambda_4 = 0$, because the network consists of four disconnected components. The remaining non-zero eigenvalues are organized into four closely-spaced groups, with gaps between these groups of non-zero eigenvalues. In contrast, if we diagonalize the graph Laplacian $L$ in Fig 1(c), there would only be one zero eigenvalue $\lambda_1 = 0$. The remaining non-zero eigenvalues are organized into a band where the gap between successive eigenvalues is small. The large spectral gap encountered during market crashes is somewhat similar to the situation shown in Fig 1(b), where we have one fully-connected cluster that is made up of two clusters $\{0, 1, 2, 9\}$ and $\{3, 4, 5, 7, 8\}$ connected through node 6 alone. The maximum spectral gap in this graph Laplacian would be associated with the narrow neck formed by node 6.

## Information cross filtering using two time windows

The use of a time window to compute cross correlations means that we have to sacrifice temporal resolution in favor of statistical significance. That is to say, if we want the cross correlations computed to be reliable, we need to use a large time window. Conversely, if we use a small time window for higher time resolution, the cross correlations computed would be less reliable. Since we are aiming to detect when the stock market switches from a normal phase to a crash phase and back, we need both reliable cross correlations and high temporal resolutions.

This can be achieved if we scan the time series using two time windows as shown in Fig 2(a). For the left window $(t_L, t_L + w)$, we associate the value of the integrated signal $I_L(t_L) = \int_{t_L}^{t_L+w} f(s)\, ds$ with its start time $t_L$. For the right window $(t_R - w, t_R)$, we associate the value of the integrated signal $I_R(t_R) = \int_{t_R-w}^{t_R} f(s)\, ds$ with its end time $t_R$. This gives us two curves $I_L(t)$ and $I_R(t)$ in Fig 2(b). In general, the integrated signal will be large over a wide range of times if a large time window is used. Therefore, the integrated signals of the left and right windows have low temporal resolutions individually.

We can think of two ways to improve the temporal resolution. The first is to multiply the two integrated signals. If we do so, the product is large only when the left and right integrated signals are simultaneously large. As shown in Fig 2(b), we find that the product signal is nonzero precisely over the duration $t_1 < t < t_2$ of the square pulse. However, this product-of-signals method can result in large spurious signals when there are no events. We illustrate this artifact in Fig 2(d). The second way to improve the temporal resolution is to multiply the derivatives of the two integrated signals as

$$-I_L'(t)I_R'(t). \tag{2}$$

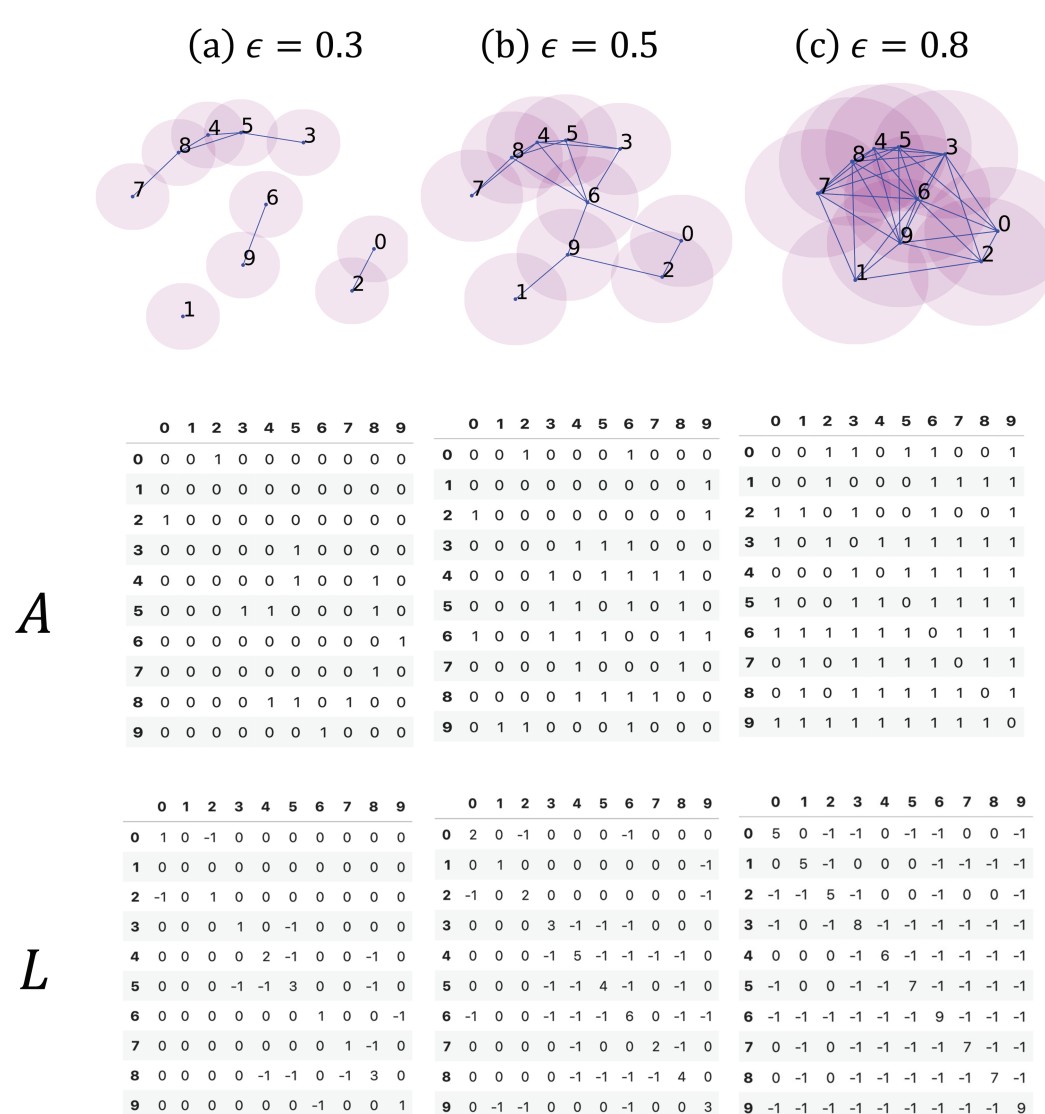

**Fig 1. Schematic figure showing how different networks (along with their adjacency matrices $A$ and graph Laplacians $L$) can be constructed from the same set of data points by varying the filtration parameter $\epsilon$.** (a) When $\epsilon = 0.3$, we find links between the nodes $(i, j) = (0, 2), (6, 9), (3, 5), (4, 5), (4, 8), (5, 8), (7, 8)$, organized into three clusters $\{0, 2\}$, $\{6, 9\}$, and $\{3, 4, 5, 7, 8\}$. The node 1 is not connected to any cluster. (b) When $\epsilon = 0.5$, all ten nodes become connected into a single cluster, with some nodes having more links than others. (c) When $\epsilon = 0.8$, some nodes become connected to all other nodes, and the network obtained is close to being a complete network. In these figures, the transparent circles centered at each data point have the same radius $\epsilon/2$. A link is drawn between two data points if their circles overlap. Below each network, we show the associated adjacency matrix $A$, whose matrix elements are $A_{i,j} = 1$ if nodes $i$ and $j$ are connected, and $A_{i,j} = 0$ otherwise. Below the adjacency matrix, we also show the graph Laplacian $L$, whose matrix elements are $L_{i,j} = \sum_j A_{i,j} - A_{i,j}$.

As shown in Fig 2(c), the product signal is a rescaled version of the square pulse. When we tested this second method in Fig 2(e), we see that there are no spurious signals. For the rest of this paper, we will perform information cross-filtering using the second method.

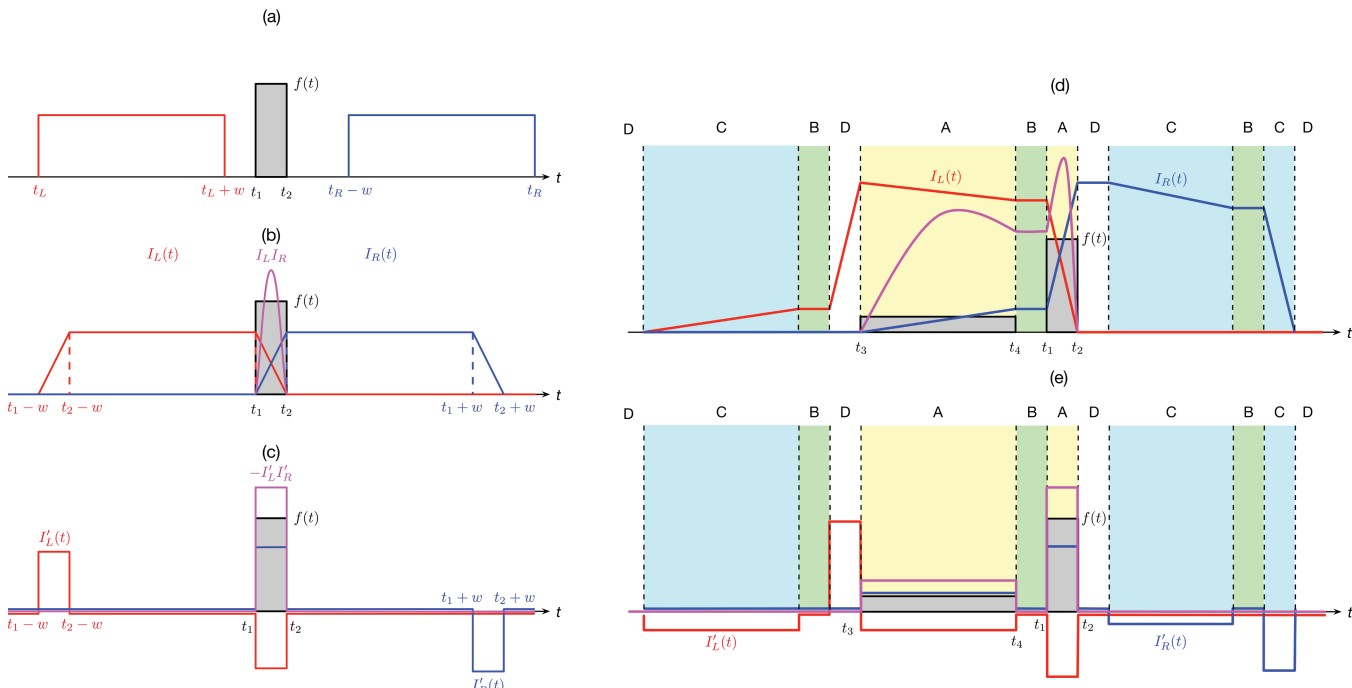

**Fig 2. (a) To accurately determine the start and end of a square pulse $f(t)$ we introduce a left time window and a right time window (both of width $w$).** (b) We plot the integrated signal obtained by the left time window at its starting time $t_L$, and then obtained by the right time window at its ending time $t_R$. We define the product signal from the two time windows as $I_L I_R$, and check that this results in a peak that starts and ends at the same time as the square pulse. (c) Another way to cross-filter the information from the two time windows is to define the product signal as $-I_L' I_R'$, where $I_L'$ and $I_R'$ are the derivatives of the left and right integrated signals. So, the product signal then has a square pulse that coincides with that in $f(t)$. (d) When $f(t)$ contains a slowly-changing signal between $t_3$ and $t_4$ in the proximity of the square pulse, a large spurious signal in $I_L I_R$ is generated not only over $(t_3, t_4)$, but also over $(t_4, t_1)$ where $f(t) = 0$. (e) If we use $-I_L' I_R'$ as the product signal, we recover only the slowly-changing signal over $(t_3, t_4)$ and the square pulse over $(t_1, t_2)$. No spurious signals are introduced.

## Results

We show the results of information cross filtering using a pair of time windows in Figs 3 and 4 for S&P 500 and Nikkei 225 whose component stocks are from developed markets (the US and Japan), as well as in Figs 5 and 6 for SGX and TWSE (both emerging markets) respectively. In these figures, we show only the product signal $-I_L' I_R'$ for time windows between two months to six months in width, increasing in steps of half a month. For the analyses presented in this section, we constructed $L_{ij}$ from $C_{ij}$ at two ultrametric distance thresholds, $\epsilon = 1.6$ and $\epsilon = 1.8$, and set maximum $\Delta\lambda$ to be the largest maximum spectral gap between these two thresholds. $I_L(t)$ is then maximum $\Delta\lambda$ over the time window $(t, t + w)$, while $I_R(t)$ is maximum $\Delta\lambda$ over the time window $(t-w, t)$.

We shall note that the cross-filtering method captures the underlying cross-correlation changes between stock market/index constituents even though some events captured may not show up in the stock market indices benchmarked in this paper. However, we expect significant market-level events like market crashes to show up both in the indices [5,29,30] and our cross-filtering method. With this in mind, let us start interpreting the results for the US market, from features that are the easiest to understand to progressively harder features. This sequence of features is thus not ordered chronologically.

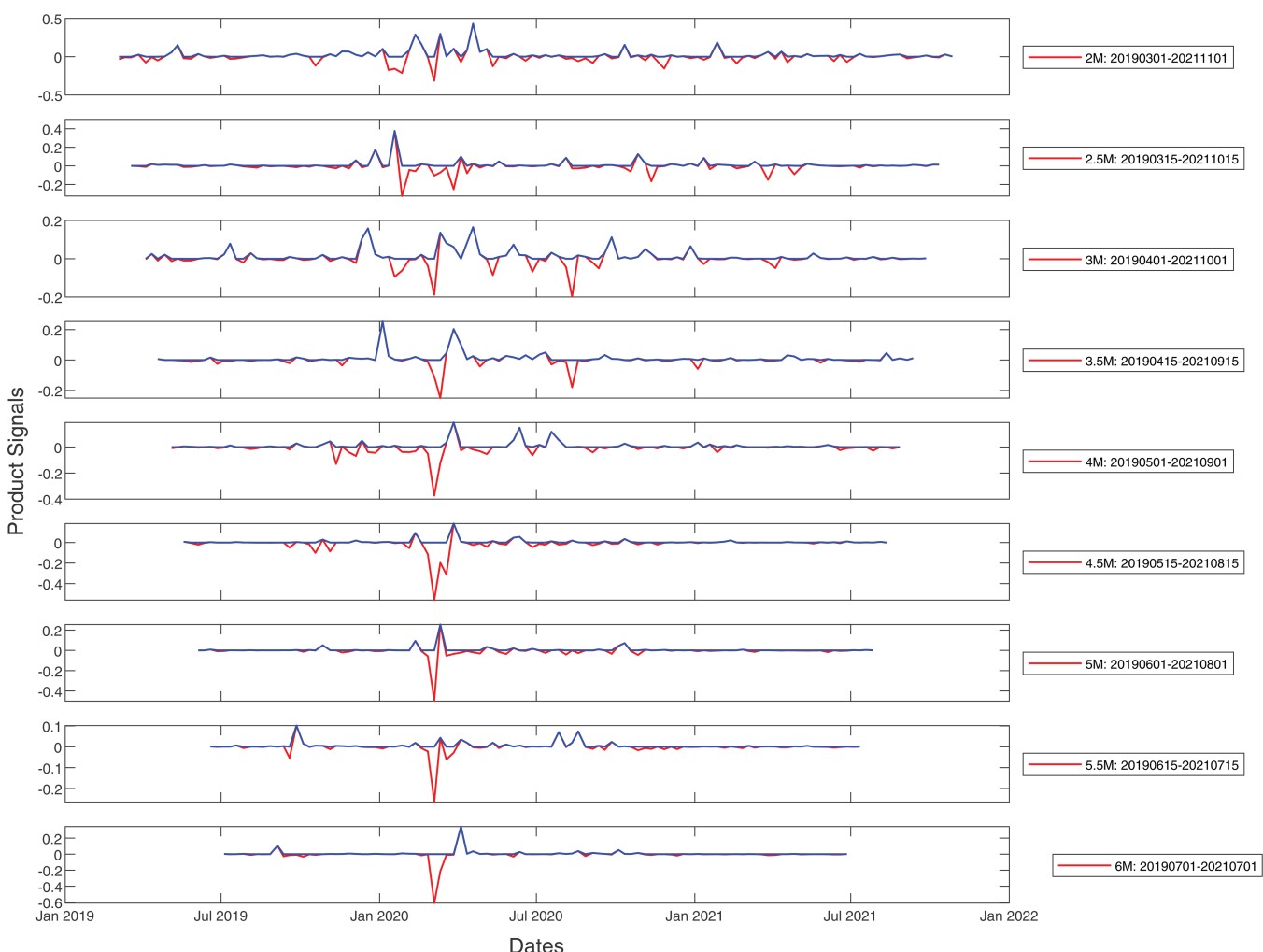

**Fig 3. Product signals of the S&P 500 for the period January 2019–January 2022, covering the March 2020 COVID-19 crash, using time window widths between two and six months.** In this plot, the red curve represents the negative product of derivatives of the left and right integrated signals with no further filtering, whereas the blue curve is obtained from the red curve after zeroing negative product signals. In terms of the blue product signals, a normal market phase corresponds to periods when the product signal is nearly zero, whereas a market crash phase is when the product signal is large and persistent.

To begin, let us discuss the peaks that appear at about the same times across all time window widths in Fig 3. We call these peaks *persistent*, because they are not sensitive to the window width used to detect them. If we look at the red curve, which is the negative product of derivatives defined in Eq (2) of the Materials and Methods section, we find both positive and negative persistent peaks. However, we do not actually expect to find negative peaks, because the maximum spectral gap indicator is always positive. We realized that negative peaks are produced by the cross-filtering method when the maximum spectral gap opens up a second or more times within the time window used. This results in the negative/positive derivative from the first event overlapping with the negative/positive derivative from the second event, to produce a negative product signal that cannot be associated with either events. Therefore, we plot the blue curve obtained by zeroing all negative product signals in the red curve. The number of persistent peaks in the blue curve is fewer than the number of persistent peaks in the red curve, but we can now be sure that all these can be associated with individual events.

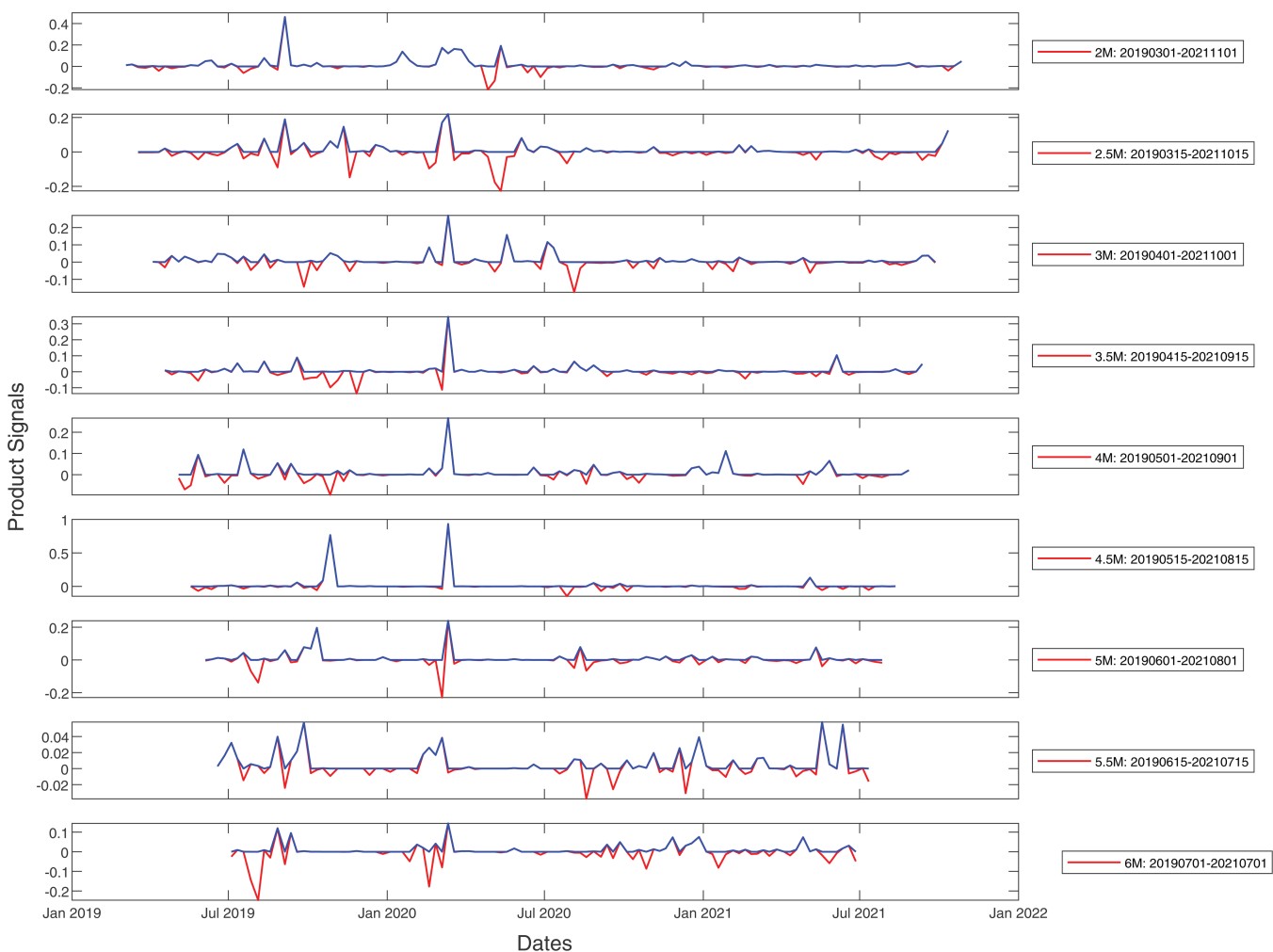

**Fig 4. Product signals of the Nikkei 225 for the period January 2019–January 2022, using time window widths between two and six months.** In this plot, the red and blue curves have the same meaning as they have in Fig 3. In terms of the blue product signals, a normal market phase corresponds to periods when the product signal is nearly zero, whereas a market crash phase is when the product signal is large and persistent.

From Fig 3, we also observe that no sizable signals are found during periods when there is no market crash.

As we can see from Fig 3, the persistent peaks in the blue product signal occurred on 11 February 2020, 11 March 2020, and 18 April 2020. Judging from the timings, this set of persistent peaks is associated with the 2020 COVID-19 market crash, which was dated to 24 February 2020 in Table 1. In Fig 3, we see that this event likely comprises three peaks, spaced approximately one month apart. This suggests that the COVID-19 crash was an extended event lasting about two months. Additionally, we find isolated peaks around September and October 2020, but only for short time windows between two and four months in widths. We call such peaks *non-persistent*, and through comparison with the S&P 500 index, realized that they were due to market corrections from the end of August 2020 to the end of October 2020.

Next, we analyze the results for Nikkei 225 shown in Fig 4. The persistent peak dated 11 March 2020 is seen across all time windows, suggesting that the COVID-19 market crash occurred around this date. For short time windows, the strength of the 11 March 2020 peak

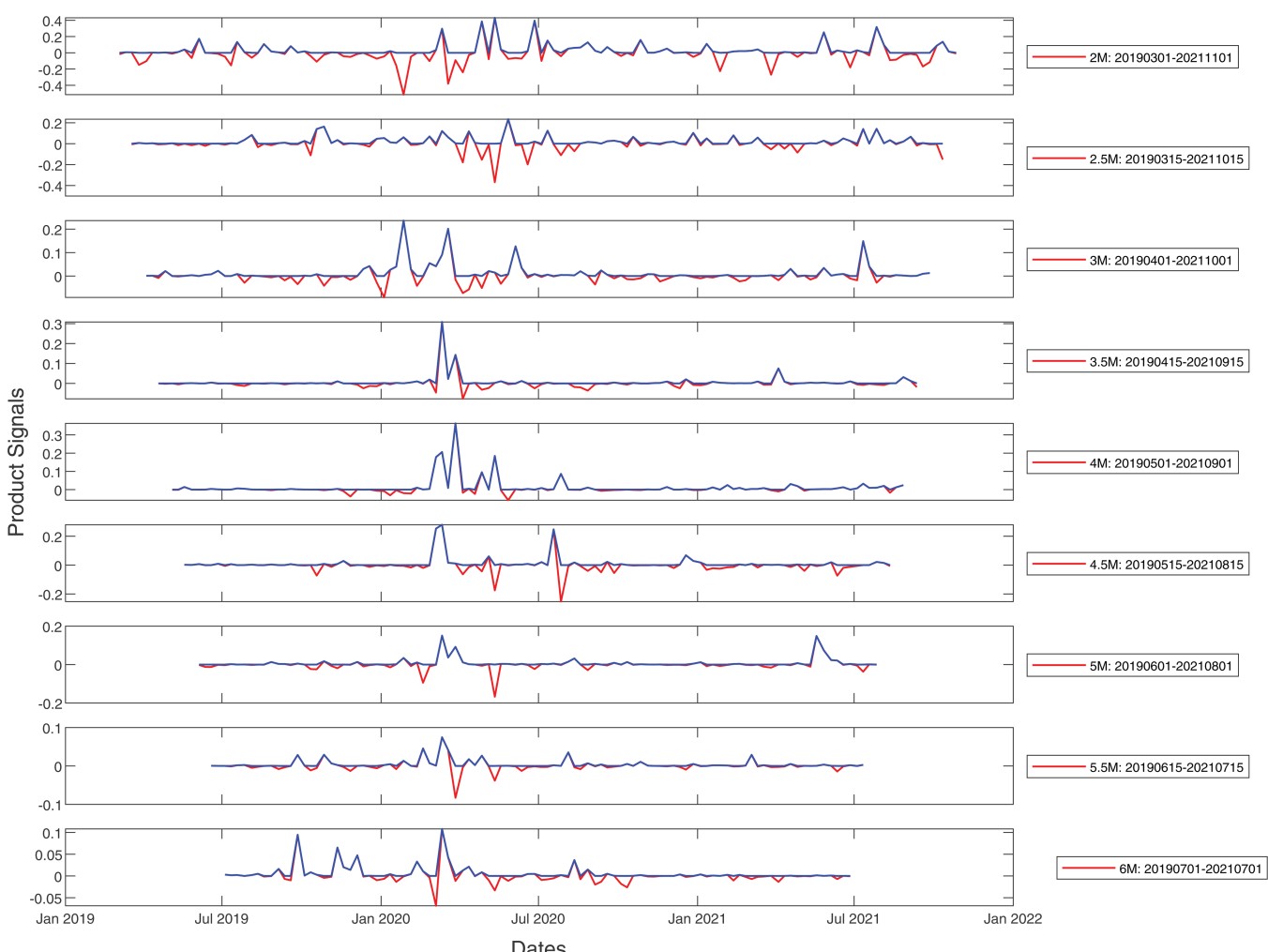

**Fig 5. Product signals of the SGX for the period January 2019–January 2022, using time window widths between two and six months.** In this plot, the red and blue curves have the same meaning as they have in Fig 3. In terms of the blue product signals, a normal market phase corresponds to periods when the product signal is nearly zero, whereas a market crash phase is when the product signal is large and persistent.

is more or less constant, but when longer time windows are used, the strength of the 11 March 2020 peak decreased, because the signal from the short market crash is diluted by the lack of signals from other parts of the time window. In contrast, the minor corrections due to the US-China trade war between July and October 2019 [86,87] and those due to strong market recovery with intermittent market corrections throughout the second half of 2020 constitute weak signals across extended periods. These appeared as periods with multiple non-persistent peaks with strengths that are largely window width independent.

Following this, we analyze the results for the SGX shown in Fig 5. Unlike the US and Japanese markets, where we find clean persistent peaks associated with the COVID-19 market crash, in the SGX we see instead an extended period of uncertainty in the form of multiple persistent peaks across all time windows from January to July 2020. The strongest peak is dated 11 March 2020 and is consistent with our prior observations in the US and Japanese markets. We also see a number of non-persistent peaks (in the 2-, 2.5-, and 3-month time windows) in the July to August 2021 period. Judging from the timing of these peaks, we

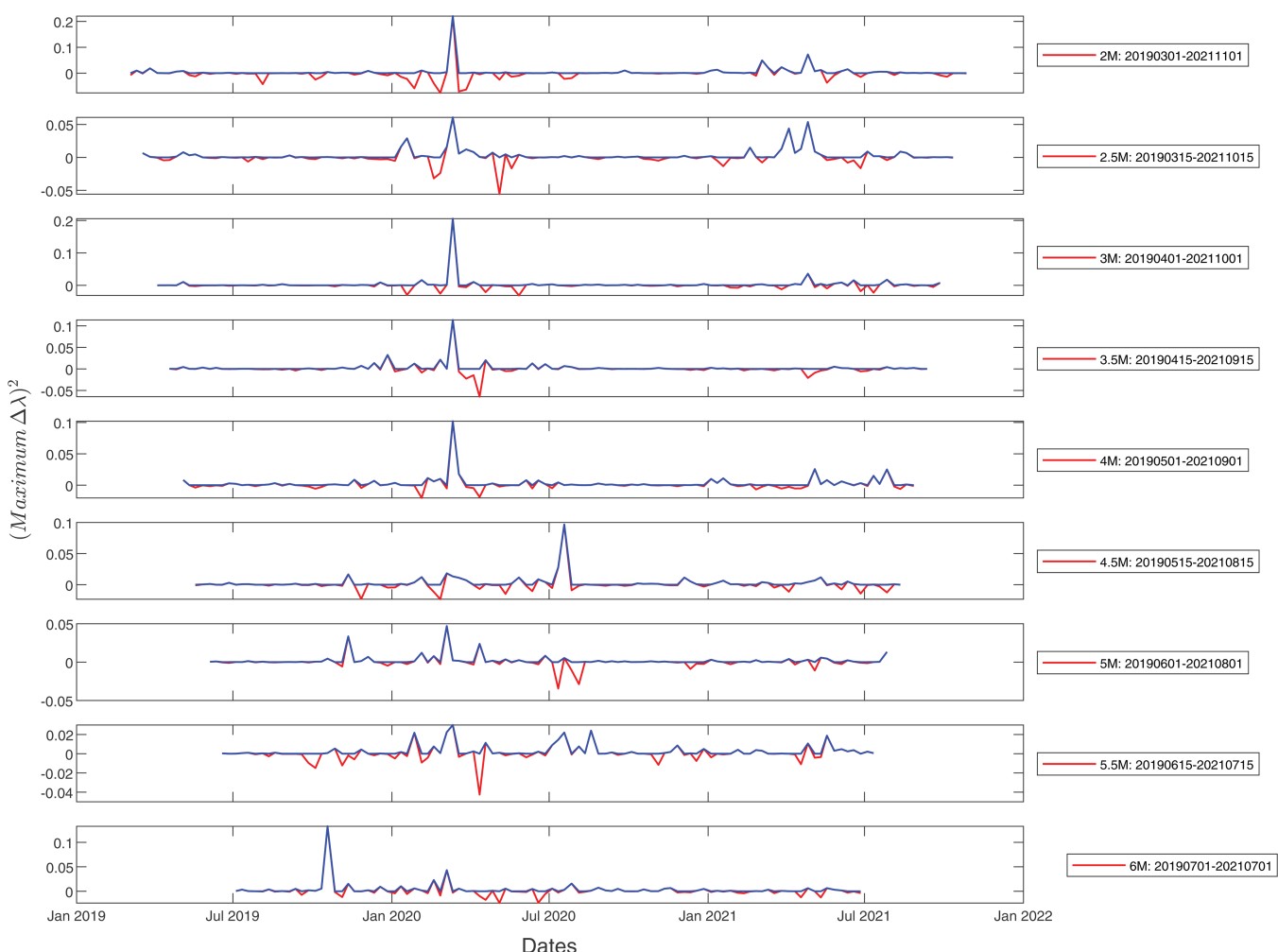

**Fig 6. Product signals of the TWSE for the period January 2019–January 2022, using time window widths between two and six months.** In this plot, the red and blue curves have the same meaning as they have in Fig 3. In terms of the blue product signals, a normal market phase corresponds to periods when the product signal is nearly zero, whereas a market crash phase is when the product signal is large and persistent. Because of the unique circumstance surrounding Taiwan, we find two highly-volatile time periods. The first highly-volatile time period contains the March 2020 COVID-19 crash. While the rest of the world came to grips with the pandemic, Taiwan reported its first local waves of infection, resulting in the second highly-volatile time period from January to July 2021.

believe they are related to the emergence of the COVID-19 delta strain [88]. We also believe that the structure of peaks is complex for SGX, because we included all stocks listed on the exchange for our analysis. During the COVID-19 period, some stocks are not traded, leading to delayed cross correlations that complicate our analysis.

Finally, let us analyze the results for TWSE shown in Fig 6. The timeline of events in TWSE is very different from S&P 500, Nikkei 225, and SGX, because Taiwan closed its borders earlier than most countries and was therefore spared the ravage by the alpha strain of COVID-19 [89]. For shorter time windows (less than 4-months), we find a single persistent peak on 11 March 2020. This is consistent with what we found for other markets. When longer time windows are used, we find a period between October 2019 (strong peak on 18 Oct 2019) and July 2020 (strong peak on 18 July 2020) where multiple non-persistent peaks emerge. We do not understand the nature of these non-persistent peaks, including the two on 18 October 2019

and 18 July 2020, as there were no significant events on these dates in the TAIEX index time series. The other interesting period with non-persistent peaks that we find in Fig 6 is between January and April 2021. We suspect that these are associated with the COVID-19 outbreak starting from a cluster of cases among China Airlines pilots improperly quarantined at the Novotel Hotel in Taoyuan County, leading eventually to community transmission in April 2021 [90]. In response, the TAIEX fell 12% over two weeks at the end of April 2021.

## Discussion

Looking back at the results we obtained for S&P 500, Nikkei 225, SGX, and TWSE, we see indeed that the strength of the information cross-filtering method is its ability to time the starts and ends of isolated events, regardless of the time window width used. Indeed, when we use the negative product of derivatives as the product signal, we have eliminated spurious signals that tend to arise when the product of integrated signals is used. Unfortunately, when the time window encompasses two or more isolated events of comparable strengths, we find the negative product signals shown in red. These we do not expect from the maximum spectral gap indicator, which is always positive. Hence, in this paper we add the condition that the product signal must be positive, and show this in blue. The price of doing so is some events missing from the product signal. As far as we can tell, this is the only limitation of our cross-filtering method. Fortunately, when we do this analysis across multiple time window widths, it becomes clear which events we missed.

To prevent ourselves from developing a biased understanding of market crashes in developed versus emerging markets based on a single market crash (March 2020 COVID-19 crash), we repeat our analysis of the S&P 500 and Nikkei 225 cross-sections of stocks for the 2008 Global Financial Crisis. Unlike for the 2020 COVID-19 pandemic, the maximum spectral gap was found at lower filtration parameters, down to $\epsilon = 1.0$. After expanding the range of $\epsilon$ to $1.0 \leq \epsilon \leq 1.8$ to search for the maximum spectral gap, we found surprisingly complex peak structures in these two markets as shown in Figs 7 and 8. There were no persistent peaks but many non-persistent ones. We suspect that this structural difference may be due to the fact that the Global Financial Crisis was endogenous, originating from the financial sector. This suggests that developed and emerging markets are not distinguished by the simple or complex persistent peak structures, but that the complexity of the peak structures is the signature of information processing by the markets, regardless of whether they are developed or emerging.

In fact, it is more compelling to think that the complex peak structure is the result of the markets digesting a series of exogenous shocks generated by government interventions. During the COVID-19 pandemic, Japan and US did not have coherent intervention measures. For example, there are no clauses in the Japanese Constitution that allowed the government to issue lockdown orders. In the US, different states introduced different measures instead of a single federal response [91–93]. In contrast, Singapore in 2020 and Taiwan in 2021 implemented nationwide lockdowns/intervention measures [90,94]. When we turn back the clock to the 2008 Global Financial Crisis, there were a lot more interventions on the part of the US government [95] and the Japanese government [96].

Finally, let us address possible comments from the change point detection community, who might object to our use of sliding windows to do change point detection. As mentioned in the Introduction section, there are three main classes of change point detection methods: (1) dynamic programming, (2) hidden Markov model, and (3) entropic segmentation method.

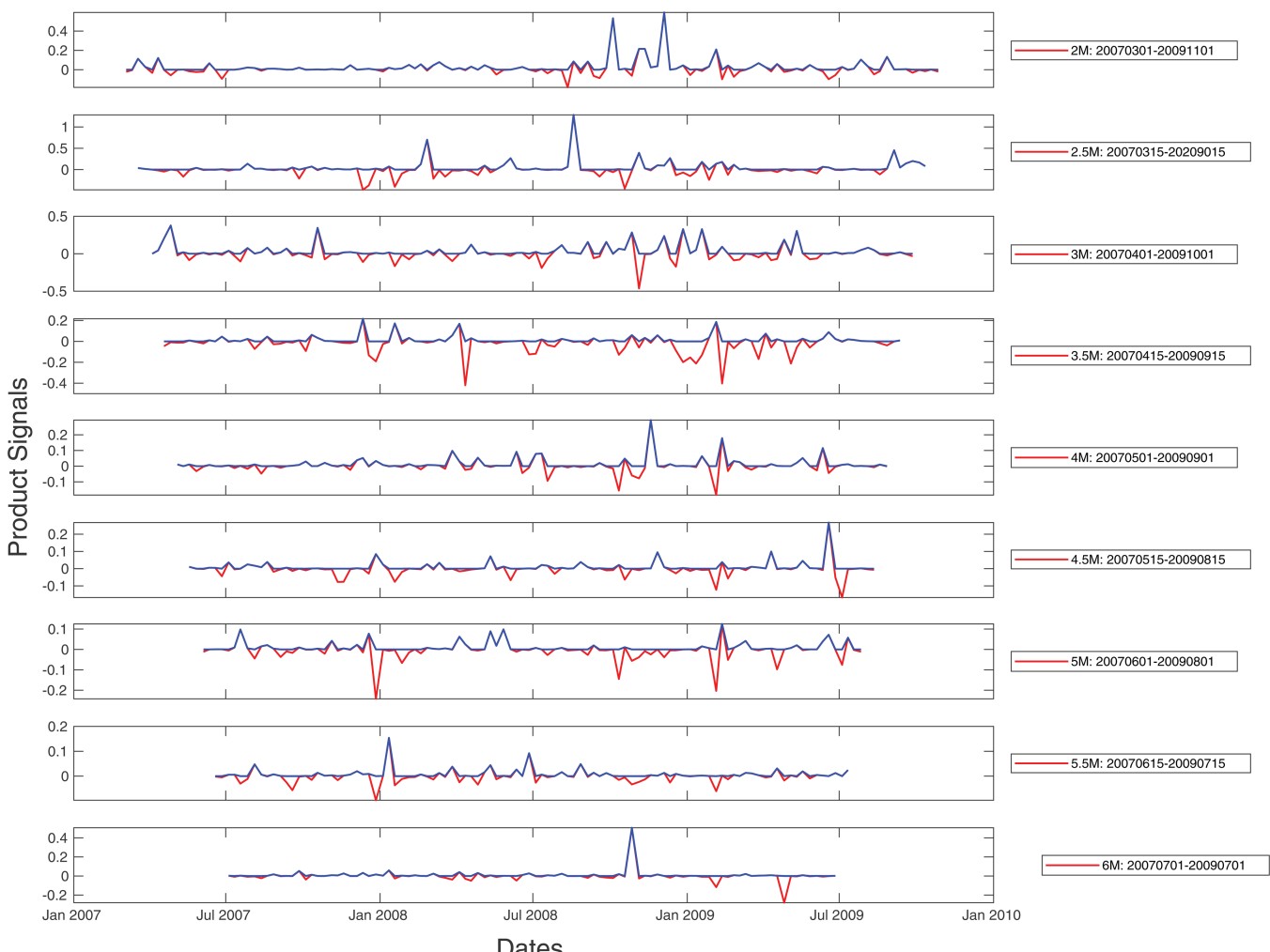

**Fig 7. Product signals of the S&P 500 for the period January 2007–January 2010, covering the Global Financial Crisis crash period, using time window widths between two and six months.** In this plot, the red curve represents the negative product of derivatives of the left and right integrated signals with no further filtering, whereas the blue curve is obtained from the red curve after zeroing negative product signals. In terms of the blue product signals, a normal market phase corresponds to periods when the product signal is nearly zero, whereas a market crash phase is when the product signal is large and persistent.

Dynamic programming is the fastest change point method, but it is designed to deal with simple statistics like mean and variances that can be computed through addition alone [37, 38]. It would be difficult to modify the dynamic programming method to deal with more complicated statistics involving spatial and temporal cross-sections, without sacrificing its speed. In particular, for our problem, the statistic that was used is the time derivative of the maximum spectral gap (maximum $\Delta\lambda$) across multiple scales $1.0 \leq \epsilon \leq 1.8$.

In comparison, the hidden Markov model method of change point detection relies on having a model of the time series data [19,20,35,36]. In general, the change points detected will not be accurate if the model is 'wrong'. When the transition probabilities are not known beforehand, and have to be estimated using various expectation maximization algorithms [97], this is also the slowest method. Entropic segmentation methods are intermediate between dynamic programming and hidden Markov model in terms of speed [39,40]. It

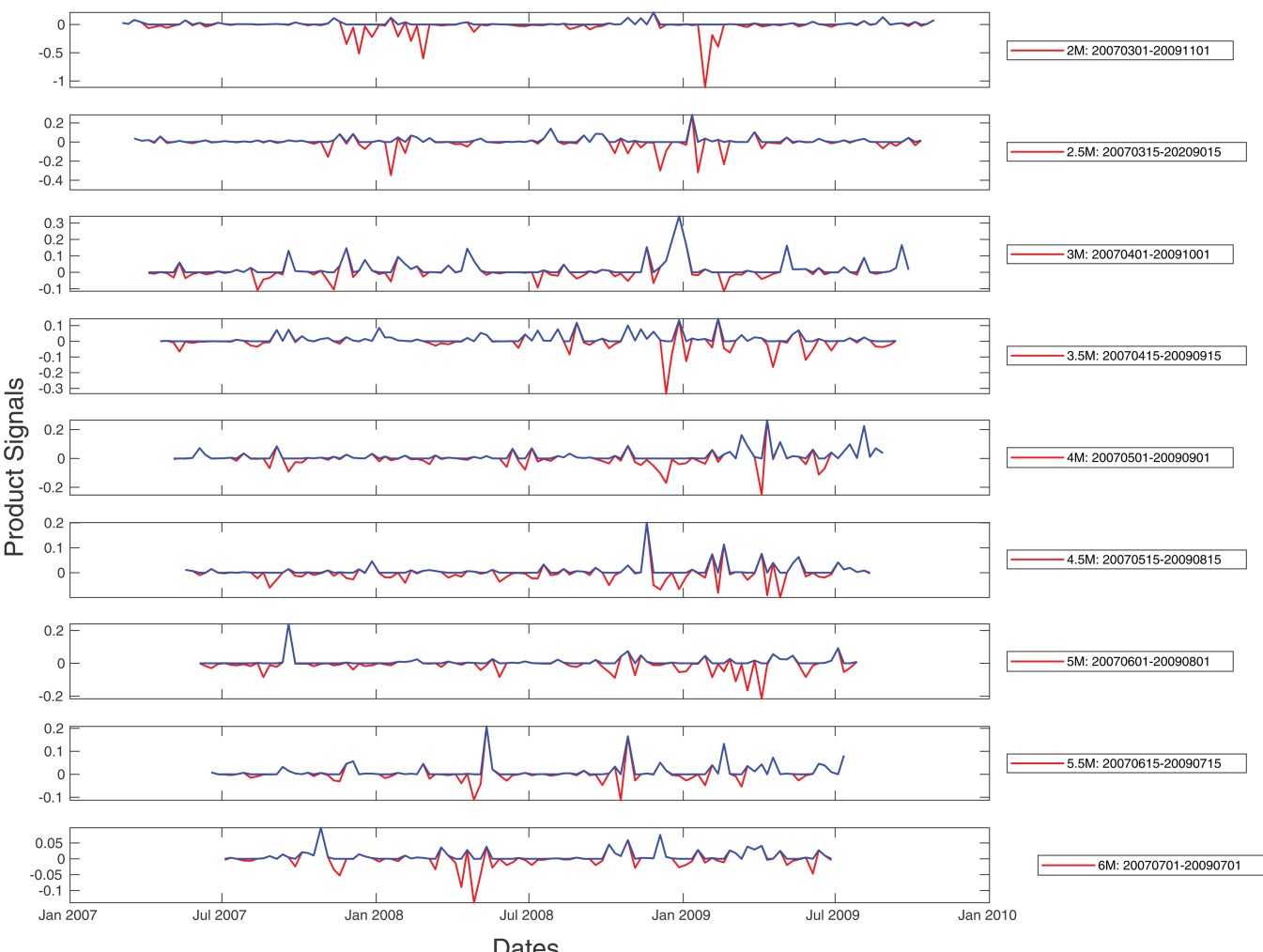

**Fig 8. Product signals of the Nikkei 225 for the period January 2007–January 2010, covering the Global Financial Crisis crash period, using time window widths between two and six months.** In this plot, the red curve represents the negative product of derivatives of the left and right integrated signals with no further filtering, whereas the blue curve is obtained from the red curve after zeroing negative product signals. In terms of the blue product signals, a normal market phase corresponds to periods when the product signal is nearly zero, whereas a market crash phase is when the product signal is large and persistent.

can be made model-independent, but this comes at a cost, i.e., discretization of continuous observed data.

All three methods are good for discontinuous changes, but will be confused by continuous changes or a mixture of continuous and discontinuous changes. In contrast, our cross-filtering method works well for both continuous, discontinuous changes and a mixture of both. Ultimately, we do not claim that the cross-filtering method presented in this paper is the last word on event detection, but that it is another useful tool to add to the suite of methods available for this purpose.

## Conclusion

In this work, we developed a method of information cross-filtering from two time windows of varying sizes to determine the start and end points of brief periods with statistics different

from the bulk of a time series. We then applied the method to analyze the time derivative of the maximum spectral gap of the graph Laplacian at scales $\epsilon = 1.6, 1.8$ from the daily returns of 460 component stocks of S&P 500, 210 components stocks of Nikkei 225, 500 stocks on the SGX, and 650 stocks on the TWSE from 1 January 2019 to 30 June 2022. The window size is varied from two to six months, in steps of half a month. We then slide these time windows by one week each time. Based on the product signals, we identified one or more persistent peaks associated with the COVID-19 market crash in the first half of 2020. For S&P 500, Nikkei 225, and SGX, we also identified non-persistent peaks, with those in the second half of 2019 suspected to be due to the US-China trade war, and those in the second half of 2021 to be associated with the delta strain of COVID-19. In contrast, the non-persistent peaks in the TWSE were due to community transmission of COVID-19 starting from Novotel Hotel.

By applying the information cross-filtering method on the four stock markets, we answered our primary research question and identified the market crash period associated with the start of the COVID-19 pandemic, in the form of sharp spikes in the cross-filtered product signals around 11 March 2020 in the US and Japanese markets. For the Singapore and Taiwan markets, the complex peak structures around this same period made it difficult for us to identify clear starts and ends to individual market crashes. This was also the case for the US and Japanese markets during the Global Financial Crisis in 2008. In relation to our secondary research question, we also realized that the main observable differences between markets were in how they digested the series of exogenous shocks, instead of them being developed or emerging markets.

Through this work, we add evidence to the understanding that market crashes are not point events, but brief periods (one week to two months) in the time series of financial markets. In the future, we hope to combine the information cross-filtering with higher order Hodge Laplacians to obtain deeper insights into market crashes and other extreme events, as well as a systematic study over multiple crashes in multiple markets. We believe information on the starts and ends of market crashes are useful in the field of risk management. Unlike high frequency traders who frequently make the most profits during market crashes, long-term investors need to hold on to their investment portfolio for an extended period of time. For these investors, the key decisive criterion would be the long-term growth prospect of the stocks they are holding on to. During a market crash, the prices of these stocks may deviate significantly from their fundamental values, making it risky to buy or sell such stocks for the purpose of portfolio re-balancing. Long-term traders such as these should wait out the market crash, examine the changes in fundamental value, if any, before resuming trading activities. This is one specific way that we can use knowledge of starts and ends of market crashes to manage our trading risk. Other risk management strategies might also be possible with this knowledge, but we shall not describe them here.

## Acknowledgments

S.A.C., Peter T.-W. Y., and Z.T.K. acknowledge Kelin Xia, Lock Yue Chew, and Charlie Charoenwong for discussions and constructive comments.

## Author contributions

**Conceptualization:** Siew Ann Cheong.

**Data curation:** Zheng Tien Kang, Peter Tsung-Wen Yen.

**Formal analysis:** Zheng Tien Kang, Peter Tsung-Wen Yen, Siew Ann Cheong.

**Investigation:** Zheng Tien Kang, Peter Tsung-Wen Yen, Siew Ann Cheong.

**Methodology:** Zheng Tien Kang, Peter Tsung-Wen Yen, Siew Ann Cheong.

**Project administration:** Siew Ann Cheong.

**Resources:** Siew Ann Cheong.

**Software:** Zheng Tien Kang, Peter Tsung-Wen Yen.

**Supervision:** Siew Ann Cheong.

**Validation:** Siew Ann Cheong.

**Writing – review & editing:** Zheng Tien Kang, Peter Tsung-Wen Yen, Siew Ann Cheong.

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
