## [Editor Report · Decision Letter 0]

12 Jun 2024

PONE-D-24-21514

Indicator from the graph Laplacian of stock market time series cross sections can precisely determine the durations of market crashes

PLOS ONE

Dear Dr. Cheong,

Thank you for submitting your manuscript to PLOS ONE. After careful consideration, we have decided that your manuscript does not meet our criteria for publication and must therefore be rejected.

I am sorry that we cannot be more positive on this occasion, but hope that you appreciate the reasons for this decision.

Kind regards,

Junhuan Zhang, PhD

Academic Editor

PLOS ONE

**Additional Editor Comments:**

The methodology and results of this paper are too simple, and the contributions of this paper to the literature are insufficient.

- - - - -

---

## [Author Response · Author response to Decision Letter 1]

18 Sep 2024

We have made substantial changes to the method and the manuscript. Please see our Response to Reviewers.

---

## [Decision Letter · Decision Letter 1]

30 Oct 2024

PONE-D-24-21514R1Indicator from the graph Laplacian of stock market time series cross sections can precisely determine the durations of market crashesPLOS ONE

Dear Dr. Cheong,

Thank you for submitting your manuscript to PLOS ONE. After careful consideration, we feel that it has merit but does not fully meet PLOS ONE’s publication criteria as it currently stands. Therefore, we invite you to submit a revised version of the manuscript that addresses the points raised during the review process.

We look forward to receiving your revised manuscript.

Kind regards,

Feier Chen, Ph.D

Academic Editor

PLOS ONE

Journal Requirements:

3. In your Methods section, please include additional information about your dataset and ensure that you have included a statement specifying whether the collection and analysis method complied with the terms and conditions for the source of the data 

Additional Editor Comments (if provided):

Reviewers' comments:

Reviewer's Responses to Questions

**Comments to the Author**

1. If the authors have adequately addressed your comments raised in a previous round of review and you feel that this manuscript is now acceptable for publication, you may indicate that here to bypass the “Comments to the Author” section, enter your conflict of interest statement in the “Confidential to Editor” section, and submit your "Accept" recommendation.

Reviewer #1: (No Response)

Reviewer #2: All comments have been addressed

Reviewer #3: (No Response)

Reviewer #4: All comments have been addressed

2. Is the manuscript technically sound, and do the data support the conclusions?

Reviewer #1: Partly

Reviewer #2: Yes

Reviewer #3: Partly

Reviewer #4: Yes

3. Has the statistical analysis been performed appropriately and rigorously? 

Reviewer #1: Yes

Reviewer #2: Yes

Reviewer #3: N/A

Reviewer #4: Yes

4. Have the authors made all data underlying the findings in their manuscript fully available?

Reviewer #1: Yes

Reviewer #2: Yes

Reviewer #3: (No Response)

Reviewer #4: Yes

5. Is the manuscript presented in an intelligible fashion and written in standard English?

Reviewer #1: No

Reviewer #2: Yes

Reviewer #3: No

Reviewer #4: Yes

6. Review Comments to the Author

Reviewer #1: The researchers explained in the Data section that they chose the four financial markets because of their knowledge. This is an insufficient and unconvincing reason for academic research. No section was allocated to reviewing the literature, and this is a major weakness in the study that must be closed by reviewing several pieces of literature so that the theoretical aspect of the research is completed.

Reviewer #2: All the comments has been addressed and they have worked on all the pin point part. Its glad to see such a paper.

Reviewer #3: Review comments

The paper suggests a distinctive approach to predicting the onset and conclusion of market collapses by utilizing graph Laplacian and information cross-filtering with two-time windows. This innovative approach contributes a new viewpoint to the published literature on financial risk management. During the COVID-19 pandemic, the research examines stock market data, particularly on the S&P 500, SGX, and TWSE markets. This data is both timely and pertinent. The manuscript provides a comprehensive explanation of the relationship between market collapses and spectral gaps, as well as well-labeled graphs for the results. The discourse regarding persistent and non-persistent peaks is insightful and provides valuable insights into comprehending market collapses.

Major Issues

1. The introduction section must be revised to explicitly state the research void, the purpose of the study, and the actual contributions that your study made.

2. Adding a literature review section and theoretical foundation section could benefit a more precise understanding.

3. Although the method is theoretically sound, it may need to be more complex for non-technical financial analysts or risk managers to employ. Cross-filtering with two-time windows and spectral gaps necessitates substantial computational resources and specialized knowledge, which may restrict its practical application beyond academic circles.

3. The paper concentrates on a significant crash event (COVID-19) in only three stock markets (the S&P 500, SGX, and TWSE). Although these markets offer some contrast, a more comprehensive analysis encompassing a wider range of markets or distinct crash events could enhance the generalisability of the results. Additionally, the proposed method's performance in non-crash market conditions must be adequately addressed, which is essential for evaluating the method's reliability in various market environments. The analysis should be expanded to encompass multiple crash events, such as the 2008 financial crisis, and should consider a more diverse range of markets (e.g., emerging vs. developed, various geographies).

4. The authors recognize the issue of "spurious peaks" that arise when closely timed events overlap, complicating the interpretation of the results. Although this is addressed in the discussion, a more comprehensive examination of the methods to mitigate this issue is necessary. Proposing a more robust solution to address overlapping events could improve future iterations of the paper.

5. The paper predominantly relies on the maximum spectral gap of the graph Laplacian to signal market crashes. Although this is innovative, the complete complexity of financial markets may need to be adequately captured by relying on a singular indicator. The accuracy and robustness of predictions may be enhanced by integrating this with other statistical or machine learning-based methods. Provide additional details regarding the practical application of this method in real-time risk management. What are the essential computational instruments, and how can financial institutions incorporate these discoveries?

6. Despite the authors' assertion that defining market collapses as durations (rather than point events) is beneficial for risk management, they do not explore the practical implications of this approach for financial institutions. The study's relevance would be improved by a more in-depth examination of the practical implications and a comparison of this method with existing financial risk management tools. Expand the discussion on non-crash periods and propose potential remedies for the spurious peaks problem. Provide additional discussion.

7. The diction could be more lucid despite its academic nature. Readers who are not acquainted with the intricacies of graph theory or Laplacian spectra may find certain sections challenging due to their dense technical content. The paper would be more accessible to a broader audience if the language in these sections were simplified or additional context information was provided.

8. The structure of the paper needs to be revised, including the introduction, literature, methodology, results, discussion, implications, limitations, future directions, and conclusion. Another problem with this manuscript is that we need to see citations in the results and discussion sections. Citations from top academics can benefit you a lot while you are making arguments.

Reviewer #4: Review of “Indicator from the graph Laplacian of stock market time series cross sections can precisely determine the durations of market crashes”

PONE-D-24-21514R1

Accepted

This paper delivers novel theoretical conclusions

Introduction

clear

Theoretical framework

sufficiently robust.

Research design

sufficiently robust.

Descriptive Statistics

The descriptive statistics is sufficiently robust.

Conclusions, Implications, and Recommendations

sufficiently robust.

7. PLOS authors have the option to publish the peer review history of their article (what does this mean?). If published, this will include your full peer review and any attached files.

Reviewer #1: No

Reviewer #2: No

Reviewer #3: **Yes: **MUHAMMAD ARSHAD

Reviewer #4: No

---

## [Author Response · Author response to Decision Letter 2]

2 Jan 2025

Please see the Response to Reviewers document.

---

## [Decision Letter · Decision Letter 2]

5 Feb 2025

PONE-D-24-21514R2Indicator from the graph Laplacian of stock market time series cross sections can precisely determine the durations of market crashesPLOS ONE

Dear Dr. Cheong,

Thank you for submitting your manuscript to PLOS ONE. After careful consideration, we feel that it has merit but does not fully meet PLOS ONE’s publication criteria as it currently stands. Therefore, we invite you to submit a revised version of the manuscript that addresses the points raised during the review process.

We note that one or more reviewers has recommended that you cite specific previously published works. As always, we recommend that you please review and evaluate the requested works to determine whether they are relevant and should be cited. It is not a requirement to cite these works. We appreciate your attention to this request.

We look forward to receiving your revised manuscript.

Kind regards,

Feier Chen, Ph.D

Academic Editor

PLOS ONE

Reviewers' comments:

Reviewer's Responses to Questions

**Comments to the Author**

1. If the authors have adequately addressed your comments raised in a previous round of review and you feel that this manuscript is now acceptable for publication, you may indicate that here to bypass the “Comments to the Author” section, enter your conflict of interest statement in the “Confidential to Editor” section, and submit your "Accept" recommendation.

Reviewer #1: (No Response)

Reviewer #5: (No Response)

Reviewer #6: (No Response)

2. Is the manuscript technically sound, and do the data support the conclusions?

Reviewer #1: Partly

Reviewer #5: Yes

Reviewer #6: Partly

3. Has the statistical analysis been performed appropriately and rigorously? 

Reviewer #1: Yes

Reviewer #5: Yes

Reviewer #6: Yes

4. Have the authors made all data underlying the findings in their manuscript fully available?

Reviewer #1: Yes

Reviewer #5: Yes

Reviewer #6: Yes

5. Is the manuscript presented in an intelligible fashion and written in standard English?

Reviewer #1: No

Reviewer #5: Yes

Reviewer #6: Yes

6. Review Comments to the Author

Reviewer #1: The manuscript demonstrates significant improvements in addressing the previous comments; however, there are still areas that require minor revisions to enhance clarity and coherence. While the manuscript is technically sound overall, the authors should provide further elaboration on the choice and justification of the selected markets to strengthen the theoretical foundation and address potential gaps in the literature. Additionally, the manuscript contains sections that are unclear or written in a way that hinders readability, and revisions are recommended to improve sentence structure, fix minor grammatical errors, and enhance the flow of ideas. Overall, I recommend Minor Revisions to address these points and enhance the overall quality of the manuscript.

Reviewer #5: Predicting the patterns of the stock market is always an important task. This paper presents an interesting perspective that is a valuable addition to the literature. One thing I feel the authors can add is the comparison with other existing methods/models. Even if it cannot be done and included in the current manuscript, it should be reviewed and included as future work. Some of the related methods can be seen below:

1. Shu M, Zhu W. (2022). Dissecting the 2015 Chinese Stock Market Crash. Stat; https://doi.org/ 10.1002/sta4.460

2. Shu M, Song R, Zhu W. (2021). The ‘COVID’Crash of the 2020 US Stock Market. The North American Journal of Economics and Finance, 101497.

Reviewer #6: Thanks for giving me an opportunity to review the manuscript entitled “Indicator from the graph Laplacian of stock market time series cross sections can precisely determine the durations of market crashes”. This manuscript proposed the information cross-filtering method to determine the start and end points of market crashes. I am concerned that the validation of method is lack of convincing, as only the COVID-19 crash was analyzed. The COVID-19 crash is a unique case of a crash that differs significantly from previous well-known market crashes, such as 2000 stock market crash. It is recommended to use different historical crash events to validate this method.

7. PLOS authors have the option to publish the peer review history of their article (what does this mean?). If published, this will include your full peer review and any attached files.

Reviewer #1: No

Reviewer #5: No

Reviewer #6: No

---

## [Author Response · Author response to Decision Letter 3]

4 Mar 2025

Editor: We note that one or more reviewers has recommended that you cite specific previously published works. As always, we recommend that you please review and evaluate the requested works to determine whether they are relevant and should be cited. It is not a requirement to cite these works. We appreciate your attention to this request.

Reply: We understand that we are not obliged to cite the works recommended by the reviewer. However, we have first-hand experience on how a paper on market crashes naturally attract questions on their predictions. Therefore, we included two additional paragraphs on predicting market crashes in the Literature Survey, thereafter it becomes natural to cite the two suggested references, along with others.

Reviewer #1:

Comment: The manuscript demonstrates significant improvements in addressing the previous comments; however, there are still areas that require minor revisions to enhance clarity and coherence.

Reply: We thank Reviewer 1 for the positive comment.

Comment: While the manuscript is technically sound overall, the authors should provide further elaboration on the choice and justification of the selected markets to strengthen the theoretical foundation and address potential gaps in the literature.

Reply: We thank Reviewer 1 for the constructive comment. MSCI, which is an authoritative source in the finance industry, lists Hong Kong and Singapore as developed markets [1] (see pages 4 and 5 of the table in the link). However, we disagree with the classification of Singapore because traders on SGX don’t have the same level of sophistication as those in NYSE and the Tokyo Stock Exchange (TSE). Also, company valuations on the SGX are lower than similar companies listed on other markets. Overall, it is a much more illiquid market compared to the former two and has seen substantial delisting/privatization far outpacing new IPOs since the lates 2010s [2–6]. Our assessment of Singapore as an emerging market is corroborated by a recent study, showing that Singapore shares similar emerging markets attributes with other developing ASEAN markets [10].

We also reviewed works from various economics and finance journals and noted that different researchers have different opinions on the taxonomy of stock markets. Firstly, through clustering and centrality analysis of the minimal spanning tree (MST) of world stock market indices, it is found that throughout 2005 to 2014 US and the developed European markets are closely linked with one another. Emerging Asian markets are further apart and linked to the developed markets via the Japanese market. The developed Japanese market, together with the US and German markets, act as hubs in the network of world stock markets [7]. A separate dynamic correlation analysis suggested that Hong Kong and Japan both exhibited developed market characteristics [8]. The US market is generally regarded as the engine of growth for the whole world, thus it is natural to include it in our studies. We also chose the Japanese market as a contrasting developed market, because it is in Asia, and because the GDP of Japan is the third largest (formerly second largest) in the world.

Having included the Japanese market as one of the two developed markets, it is natural to include emerging markets from Asia as a contrast. In particular, the Asian Tigers of Hong Kong, Singapore, South Korea, and Taiwan would be very good choices because of their rapid ascent as emerging markets. In an analysis of events unfolding during the 2008 US subprime crisis, South Korea and Singapore are considered emerging markets [9]. In another study investigating the financial interdependence between markets, South Korea and Taiwan are listed alongside Mexico as among the more advanced emerging markets [10]. However, we realized that it is not necessary to include all four markets in our study, not least because of the classification of Hong Kong as a developed market. In general, strong correlations are observed between the Hong Kong and Singapore markets, as seen during the 1997 Asian Financial Crisis [8] and their relative positions in the MST analysis of the recent COVID-19 crash [7,11,12]. Because of their similar industry mix, and focus on the semiconductor industry, we also see that Taiwan is the nearest neighbour of South Korea in the MST [7,11,12]. Therefore, we chose Singapore from the Hong Kong-Singapore pair and Taiwan from the South Korea-Taiwan pair of emerging markets as contrasts to the US and Japanese markets.

Through the literature review above, we are confident that Taiwan and Singapore are correctly classified as emerging markets (though more advanced than other emerging markets) while US and Japan are truly developed markets, in agreement with other experts.

Comment: Additionally, the manuscript contains sections that are unclear or written in a way that hinders readability, and revisions are recommended to improve sentence structure, fix minor grammatical errors, and enhance the flow of ideas. Overall, I recommend Minor Revisions to address these points and enhance the overall quality of the manuscript.

Reply: We apologize to Reviewer 1 still finds our writing difficult to follow. We have been very careful in our revision, in particular we have systematically rephrased sentences that sound ambiguous to readers. We are happy to further improve the readability of our manuscript, and therefore we have utilized several language checker tools such as ChatGPT and LanguageTool extension on Firefox to check for ambiguity and grammatical errors. The ChatGPT language checker function is integrated with Overleaf. Potential errors are underlined in yellow for us to act.

As the Overleaf built-in checker function provides limited number of corrections and requires a premium subscription for full functionality, we also tried with the free LanguageTool extension on Firefox (also available on Google Chrome) to check for errors. Suggested correction of names of people and places in both tools are ignored as they are proper nouns.

We are hopeful that with the thorough analysis of the manuscript with both language checker tools, we have addressed concerns on language issues from Reviewer 1. In case we overlooked and missed any sections, we hope to hear back from Reviewer 1 on the particular sections that hinder readability.

Reviewer #5:

Comment: Predicting the patterns of the stock market is always an important task. This paper presents an interesting perspective that is a valuable addition to the literature.

Reply: We thank Reviewer 5 for the positive comments on the works done.

Comment: One thing I feel the authors can add is the comparison with other existing methods/models. Even if it cannot be done and included in the current manuscript, it should be reviewed and included as future work. Some of the related methods can be seen below:

Shu M, Zhu W. (2022). Dissecting the 2015 Chinese Stock Market Crash. Stat; https://doi.org/10.1002/sta4.460

Shu M, Song R, Zhu W. (2021). The ‘COVID’ Crash of the 2020 US Stock Market. The North American Journal of Economics and Finance, 101497.

Reply: We thank Reviewer 5 for the suggestions. We have added two new paragraphs on market crash predictions to the Literature Review section. The first paragraph is on prediction methods involving the use of Schefferian early warning indicators (SEWIs). SEWIs include increased variance, skewness, and kurtosis, as well as critical slowing down and spectral reddening, along with others like flickering and patchiness [13–17]. In their reviews [18,19], Scheffer et al. explained the critical slowing down phenomenon originated from the underlying dynamical systems. Whenever a complex system approaches a critical point or tipping point, it will take longer to return back to its equilibrium after perturbations. This universal behaviour precedes all critical transitions, thus the SEWIs can be used to identify and predict regime shifts in financial markets. Much inspired by the existing works, we also studied the US housing market crash during 2004 to 2008 [20,21], and more recently on high-frequency FOREX time series [22], using the SEWIs.

In the second paragraph, we described the log-periodic power law (LPPL) method, which is used by many more groups for the prediction of market crashes. This method was developed independently by two groups of people, namely, Feigenbaum et al. [23,24] and Sornette et al. [25]. Through fitting the financial time series to power laws with log-periodic decorations, we can estimate how far away is the crash (or any other extreme events) from the current time, the degree of super exponential growth of the current market state and the angular frequency of the log-periodicity [26–34]. In the figure below (but not in the manuscript), we show three successful LPPL fits to the Nasdaq composite index, the S&P 500 index and the Nikkei 225 index. Please refer to the Response to Reviewers document for related plots.

It is important to note that LPPL provides insights on a single time series only. Hence, underlying cross-correlations changes among index constituents will not be captured by the LPPL method.

Reviewer #6:

Comment: Thanks for giving me an opportunity to review the manuscript entitled “Indicator from the graph Laplacian of stock market time series cross sections can precisely determine the durations of market crashes”. This manuscript proposed the information cross-filtering method to determine the start and end points of market crashes.

Reply: We thank Reviewer 6 for the review.

Comment: I am concerned that the validation of method is lack of convincing, as only the COVID-19 crash was analyzed. The COVID-19 crash is a unique case of a crash that differs significantly from previous well-known market crashes, such as 2000 stock market crash. It is recommended to use different historical crash events to validate this method.

Reply: We thank Reviewer 6 for the suggestion. Since we worked with market cross-sections (including the whole of SGX and TWSE), the heavy computational load limited out initial focus to the COVID-19 crash period. We have since performed additional analysis on the 2008 Global Financial Crisis for the S&P 500 components and Nikkei 225 components and described our findings in the Discussion section, so as to not complicate the story-telling in the Results section.

We now believe that the complex peak structures seen in Fig. 5 and Fig. 6 (for COVID-19), as well as Fig. 7 and Fig. 8 (for the Global Financial Crisis) are the results of markets digesting a sequence of exogenous shocks generated by government interventions. We believe the simple peak structures seen in Fig. 3 and Fig. 4 for COVID-19 are the consequences of a lack of coherent government responses to the pandemic. We wish to thank Reviewer 6 for suggesting us to inspect other market crashes, pointing us to this alternative but more compelling interpretation of our results. We have changed part of the Conclusion to reflect this new understanding.

References

1. Developed Markets Indexes | MSCI Indexes. [cited 16 Feb 2025]. Available: https://www.msci.com/indexes/group/developed-markets-indexes?index-sort=aToZ

2. Making the Singapore market great again. In: The Edge Singapore [Internet]. 17 Oct 2024 [cited 16 Feb 2025]. Available: https://www.theedgesingapore.com/news/markets/making-singapore-market-great-again

3. Chakraborty R. Pace of SGX delistings may not slow down in 2025: analysts. In: The Business Times [Internet]. 6 Jan 2025 [cited 16 Feb 2025]. Available: https://www.businesstimes.com.sg/companies-markets/pace-sgx-delistings-may-not-slow-down-2025-analysts

4. Ruehl M, Lockett H. Singapore exchange suffers run of delistings. Financial Times. 17 Jun 2020. Available: https://www.ft.com/content/35057281-2c33-42f5-a211-c458731f97fd. Accessed 16 Feb 2025.

5. Singapore Braces for More Delistings Even After Rule Fix. Bloomberg.com. 24 Jul 2019. Available: https://www.bloomberg.com/news/articles/2019-07-24/singapore-braces-for-delistings-to-continue-even-after-rule-fix. Accessed 16 Feb 2025.

6. The incredible shrinking Singapore stock market. The Straits Times. 12 Feb 2019. Available: https://www.straitstimes.com/business/companies-markets/the-incredible-shrinking-singapore-stock-market. Accessed 16 Feb 2025.

7. Wang G-J, Xie C, Stanley HE. Correlation Structure and Evolution of World Stock Markets: Evidence from Pearson and Partial Correlation-Based Networks. Comput Econ. 2018;51: 607–635. doi:10.1007/s10614-016-9627-7

8. Chiang TC, Jeon BN, Li H. Dynamic correlation analysis of �nancial contagion: Evidence from Asian markets.

9. Dooley M, Hutchison M. Transmission of the U.S. subprime crisis to emerging markets: Evidence on the decoupling-recoupling hypothesis.

10. Aloui R, Aïssa MSB, Nguyen DK. Global financial crisis, extreme interdependences, and contagion effects: The role of economic structure? J Bank Finance. 2011;35: 130–141. doi:10.1016/j.jbankfin.2010.07.021

11. Memon BA, Yao H. The Impact of COVID-19 on the Dynamic Topology and Network Flow of World Stock Markets. J Open Innov Technol Mark Complex. 2021;7: 241. doi:10.3390/joitmc7040241

12. Lavin JF, Valle MA, Magner NS. A Network‐Based Approach to Study Returns Synchronization of Stocks: The Case of Global Equity Markets. Xin B, editor. Complexity. 2021;2021: 7676457. doi:10.1155/2021/7676457

13. Ismail MS, Noorani MSM, Ismail M, Razak FA, Alias MA. Early warning signals of financial crises using persistent homology. Phys Stat Mech Its Appl. 2022;586: 126459. doi:10.1016/j.physa.2021.126459

14. Wang J, Zeng C, Han X, Ma Z, Zheng B. Detecting early warning signals of financial crisis in spatial endogenous credit model using patch-size distribution. Phys Stat Mech Its Appl. 2023;625: 128925. doi:10.1016/j.physa.2023.128925

15. Diks C, Hommes C, Wang J. Critical slowing down as an early warning signal for financial crises? Empir Econ. 2019;57: 1201–1228. doi:10.1007/s00181-018-1527-3

16. Song S, Li H. Early warning signals for stock market crashes: empirical and analytical insights utilizing nonlinear methods. EPJ Data Sci. 2024;13: 16. doi:10.1140/epjds/s13688-024-00457-2

17. Bertschinger N, Pfante O. Early Warning Signs of Financial Market Turmoils. J Risk Financ Manag. 2020;13: 301. doi:10.3390/jrfm13120301

18. Scheffer M, Bascompte J, Brock WA, Brovkin V, Carpenter SR, Dakos V, et al. Early-warning signals for critical transitions. Nature. 2009;461: 53–59. doi:10.1038/nature08227

19. Scheffer M, Carpenter SR, Lenton TM, Bascompte J, Brock W, Dakos V, et al. Anticipating Critical Transitions. Sci Am Assoc Adv Sci. 2012;338: 344–348. doi:10.1126/science.1225244

20. Tan J, Cheong SA. The regime shift associated with the 2004–2008 US housing market bubble. PLoS One. 2016;11: e0162140.

21. Tan JPL, Cheong SSA. Critical slowing down associated with regime shifts in the US housing market. Eur Phys J B. 2014;87: 1–10.

22. Wen H, Ciamarra MP, Cheong SA. How one might miss early warning signals of critical transitions in time series data: A systematic study of two major currency pairs. PLOS ONE. 2018;13: e0191439. doi:10.1371/journal.pone.0191439

23. FEIGENBAUM JA, FREUND PGO. DISCRETE SCALE INVARIANCE IN STOCK MARKETS BEFORE CRASHES. Int J Mod Phys B. 1996;10: 3737–3745. doi:10.1142/S021797929600204X

24. Feigenbaum JA, Freund PGO. Discrete Scale Invariance and the “Second Black Monday.” Mod Phys Lett B. 1998;12: 57–60. doi:10.1142/S0217984998000093

25. Sornette D, Johansen A, Bouchaud J-P. Stock Market Crashes, Precursors and Replicas. J Phys I. 1996;6: 167–175. doi:10.1051/jp1:1996135

26. Shu M, Song R, Zhu W. The ‘COVID’ crash of the 2020 U.S. Stock market. North Am J Econ Finance. 2021;58: 101497. doi:10.1016/j.najef.2021.101497

27. Shu M, Zhu W. Dissecting the 2015 Chinese stock market crash. Stat. 2022;11: e460. doi:10.1002/sta4.460

28. Song R, Shu M, Zhu W. The 2020 global stock market crash: Endogenous or exogenous? Phys Stat Mech Its Appl. 2022;585: 126425. doi:10.1016/j.physa.2021.126425

29. Shu M, Zhu W. Detection of Chinese stock market bubbles with LPPLS confidence indicator. Phys Stat Mech Its Appl. 2020;557: 124892. doi:10.1016/j.physa.2020.12

---

## [Decision Letter · Decision Letter 3]

13 May 2025

PONE-D-24-21514R3Indicator from the graph Laplacian of stock market time series cross sections can precisely determine the durations of market crashesPLOS ONE

Dear Dr. Cheong,

Thank you for submitting your manuscript to PLOS ONE. After careful consideration, we feel that it has merit but does not fully meet PLOS ONE’s publication criteria as it currently stands. Therefore, we invite you to submit a revised version of the manuscript that addresses the points raised during the review process.

We note that one or more reviewers has recommended that you cite specific previously published works. As always, we recommend that you please review and evaluate the requested works to determine whether they are relevant and should be cited. It is not a requirement to cite these works. We appreciate your attention to this request.

We look forward to receiving your revised manuscript.

Kind regards,

Feier Chen, Ph.D

Academic Editor

PLOS ONE

Journal Requirements:

Reviewers' comments:

Reviewer's Responses to Questions

**Comments to the Author**

1. If the authors have adequately addressed your comments raised in a previous round of review and you feel that this manuscript is now acceptable for publication, you may indicate that here to bypass the “Comments to the Author” section, enter your conflict of interest statement in the “Confidential to Editor” section, and submit your "Accept" recommendation.

Reviewer #1: (No Response)

Reviewer #5: All comments have been addressed

Reviewer #7: (No Response)

2. Is the manuscript technically sound, and do the data support the conclusions?

Reviewer #1: Yes

Reviewer #5: Yes

Reviewer #7: Yes

3. Has the statistical analysis been performed appropriately and rigorously? 

Reviewer #1: Yes

Reviewer #5: Yes

Reviewer #7: Yes

4. Have the authors made all data underlying the findings in their manuscript fully available?

Reviewer #1: Yes

Reviewer #5: Yes

Reviewer #7: Yes

5. Is the manuscript presented in an intelligible fashion and written in standard English?

Reviewer #1: Yes

Reviewer #5: Yes

Reviewer #7: Yes

6. Review Comments to the Author

Reviewer #1: Thank you for your revisions. Your manuscript now demonstrates a coherent structure and a robust statistical approach that strongly supports the conclusions. The selection and explanation of your chosen markets is clear, and your writing style is sufficiently polished for publication. Overall, this version appears ready for acceptance.

Reviewer #5: I am satisfied with the review.

The authors have thoroughly addressed my questions.

I recommend publication.

Reviewer #7: I enjoyed the manuscript. The cross filtering is interesting, and I learned something new about the max spectral gap indicator. I have some minor points that should be addressed before publication.

1. "It is widely believed that daily market returns are normally distributed, with a mean close to zero, and standard deviation between 1%–2%" I would not say that this is still widely believed. I would say, "a common assumption", or something of the sort. I think most know returns are heavy tailed

2. Literature review: The nonparametric multivariate changepoint literature, especially for changes in covariance should be at least pointed to, these methods can handle multiple series at once and some don't rely on parametric models. see e.g.,

- Matteson, David S., and Nicholas A. James. "A nonparametric approach for multiple change point analysis of multivariate data." Journal of the American Statistical Association 109.505 (2014): 334-345.

- Kelly Ramsay, Shojaeddin Chenouri, Robust nonparametric multiple changepoint detection for multivariate variability,

Econometrics and Statistics, 2023,ISSN 2452-3062, https://doi.org/10.1016/j.ecosta.2023.09.001.

3. "Second, some data may include ‘NaN’s, so we replaced them with ‘0’s." You should not replace the prices with zeros - 0 is not a reasonable number for a stock price, and this artificially reduces the variance. At least replace with the previous day's price, or something more realistic

4. "which is the negative product of slopes defined in the Materials and Methods section, we find both positive and negative persistent peaks." By slopes do you mean derivatives? The word slope does not appear before this. You should use consistent language. This is not clear. Put an equation number in Methods section and reference that.

7. PLOS authors have the option to publish the peer review history of their article (what does this mean?). If published, this will include your full peer review and any attached files.

Reviewer #1: No

Reviewer #5: **Yes: **Wei Zhu

Reviewer #7: No

---

## [Author Response · Author response to Decision Letter 4]

28 May 2025

Please see Response to Reviewers attached.

---

## [Decision Letter · Decision Letter 4]

15 Jun 2025

Indicator from the graph Laplacian of stock market time series cross sections can precisely determine the durations of market crashes

PONE-D-24-21514R4

Dear Dr. Cheong,

We’re pleased to inform you that your manuscript has been judged scientifically suitable for publication and will be formally accepted for publication once it meets all outstanding technical requirements.

Kind regards,

Feier Chen, Ph.D

Academic Editor

PLOS ONE

Additional Editor Comments (optional):

Reviewers' comments:

Reviewer's Responses to Questions

**Comments to the Author**

1. If the authors have adequately addressed your comments raised in a previous round of review and you feel that this manuscript is now acceptable for publication, you may indicate that here to bypass the “Comments to the Author” section, enter your conflict of interest statement in the “Confidential to Editor” section, and submit your "Accept" recommendation.

Reviewer #7: All comments have been addressed

2. Is the manuscript technically sound, and do the data support the conclusions?

Reviewer #7: Yes

3. Has the statistical analysis been performed appropriately and rigorously? 

Reviewer #7: Yes

4. Have the authors made all data underlying the findings in their manuscript fully available?

Reviewer #7: Yes

5. Is the manuscript presented in an intelligible fashion and written in standard English?

Reviewer #7: Yes

6. Review Comments to the Author

Reviewer #7: Thank you for addressing the comments. I'd probably have just trimmed the zeros off of the beginning and the end of the series , and just use a reduced series, but this is fine.

7. PLOS authors have the option to publish the peer review history of their article (what does this mean?). If published, this will include your full peer review and any attached files.

Reviewer #7: No

---

## [Editor Report · Acceptance letter]

PONE-D-24-21514R4

PLOS ONE

Dear Dr. Cheong,

I'm pleased to inform you that your manuscript has been deemed suitable for publication in PLOS ONE. Congratulations! Your manuscript is now being handed over to our production team.

Kind regards,

on behalf of

Dr. Feier Chen

Academic Editor

PLOS ONE